# LOW-COST ENHANCER FOR TEXT ATTRIBUTED GRAPH LEARNING VIA GRAPH ALIGNMENT

## ABSTRACT

Many graphs can be represented as Text-attributed Graphs (TAGs). Due to the rich textual information present in each node of TAGs, traditional graph neural networks (GNNs) often struggle to deliver satisfactory performance. Recent advancements leveraging large language models (LLMs) to augment new node text features have notably enhanced node representations, resulting in significant performance improvements. However, these methods typically require extensive annotations or fine-tuning on all nodes, which are both time-consuming and expensive. To address this challenge, we propose GAGA, a novel and lightweight framework for TAG representation learning. GAGA employs a more efficient strategy by annotating only representative nodes and edges, thereby reducing both annotation time and cost. It further capitalizes on these annotations by constructing an annotation graph that captures the topological relationships among them. Additionally, GAGA introduces a two-level alignment module to integrate the annotation graph with the TAG, ensuring effective alignment of their underlying structures. Experiments demonstrate that GAGA achieves classification accuracies comparable to or exceeding state-of-the-art methods while requiring only 1% of the data to be annotated, making it highly efficient.

## 1 INTRODUCTION

In real-world scenarios, many graphs can be effectively represented as text-attributed graphs (TAGs) (Yang et al., 2021), such as paper citation networks and product purchase graphs. In TAGs, nodes represent pieces of text, and edges signify connections between these nodes, such as citations or purchases. A classic example is the Ogbn-arxiv (Hu et al., 2020a) dataset, where nodes represent the titles and abstracts of academic papers, and edges denote citation relationships. TAGs hold significant practical value in areas such as text classification (Yang et al., 2015), recommendation systems (Juan et al., 2023), social networks (Li et al., 2022), and fake news detection (Kananian et al., 2024).

In recent years, numerous models have been proposed to tackle challenges related to node classification (Maekawa et al., 2022) and link prediction (Zhang & Chen, 2018) in TAGs. For example, traditional graph neural networks (GNNs) like Graph Convolutional Network (GCN) (Kipf & Welling, 2016a) and GraphSAGE (Hamilton et al., 2017) have been widely used for node classification tasks on TAGs. However, compared to classical graph data, directly leveraging these GNN architectures often struggles to achieve satisfactory performance on TAGs, particularly on large-scale datasets like Ogbn-arxiv. A key challenge with TAGs is that canonical GNN models tend to emphasize the graph structure while potentially underutilizing the rich textual information associated with each node, which is essential for capturing the context of nodes (Yan et al., 2023).

To further improve the performance, some recent works have started to augment information on graph data. Specifically, with the advent of large language models (LLMs), several studies started to leverage LLMs to generate new textual features for TAGs. The primary motivation for incorporating LLM-generated textual features is to utilize the vast world knowledge embedded in these models. This allows for the extraction of high-quality features that surpass those derived from raw text alone (He et al., 2023; Duan et al., 2023). For example, TAPE (He et al., 2023) prompts LLMs to predict node categories and to provide explanations for their own predictions. These predictions and explanations can be seen as summarization or an enhancer for each node and utilized as new

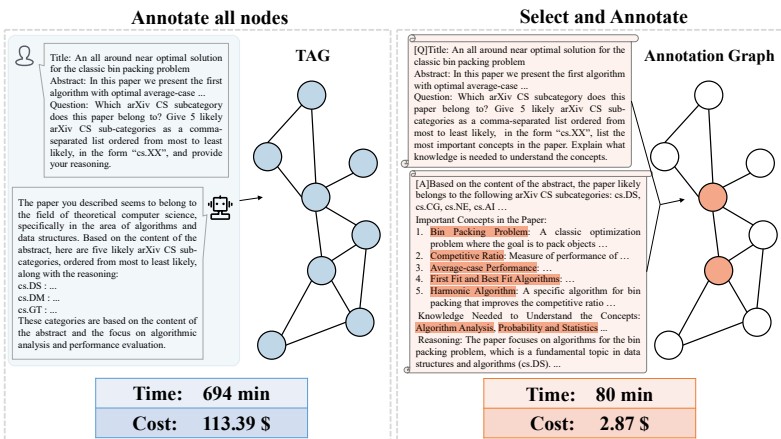

Figure 1: Comparison of TAPE and GAGA. Unlike TAPE, GAGA does not annotate all nodes. Instead, it employs representative selection to identify and annotate only the most important nodes. This approach significantly reduces both the time and cost associated with annotating.

textual features. Sun et al. (2023) takes a different approach by using LLMs to refine the topological structure of TAGs for node classification. This method begins by employing LLMs to assess semantic similarity between node attributes, which guides the addition and deletion of edges to refine the graph structure. Furthermore, pseudo-labels generated by the LLMs are used for pseudo-label propagation, acting as regularization to guide the GNNs to learn appropriate edge weights. Other work, such as SIMTEG (Duan et al., 2023), first fine-tune language models using node classification and link prediction tasks to learn node representations. Subsequently, it trains GNNs based on the embeddings generated by the LLM for node classification. While this approach is straightforward, fine-tuning LLMs can be time-consuming when dealing with large datasets.

Although existing literature leveraging the powerful text-processing capabilities of LLMs has achieved impressive results in classification tasks, several critical limitations remain (see Figure 1 for an illustration). Firstly, current approaches need to generate summarization or refine features for each node or edge, which can be highly time-consuming, especially when dealing with large datasets. For example, as noted by He et al. (2023), generating predictions and explanations for the Ogbn-arxiv dataset takes about 9 hours. Secondly, after annotating or refining, previous methods need to fine-tune all data, making them further time-consuming. Thirdly, using LLMs for annotation is expensive. Existing methods require LLMs to annotate each node, which incurs extremely high costs and is impractical for large datasets. For example, TAPE costs approximately $128 for Ogbn-arxiv.

To address the limitations outlined before, we propose **GAGA**: Graph Alignment Guided Text-attributed Graph Learning. Instead of generating summarization or refining features for each node, GAGA focuses on a more efficient strategy by only annotating representative nodes (edges), thereby reducing both the time and cost of annotation via LLMs. To further utilize these annotations, GAGA recognizes and incorporates the topological relationships between them by constructing an annotation graph. Then, an alignment module between the annotation graph and the TAGs is introduced, ensuring that the underlying structures of both graphs are effectively integrated.

There are three stages in GAGA. We aim to obtain a representative set of nodes or edges in the first stage. Then, we generate annotations for these nodes or edges by prompting LLMs. These annotations are constructed into an annotation graph. In the second stage, through contrastive learning between these sub-annotation graphs and their corresponding sub-textual graphs, we simultaneously fine-tune an LLM and a GNN. During our novel-designed and lightweight two-level contrastive learning process, the LLM captures semantic information from the text, while the GNN processes the topological structure information of the annotation graphs and textual graphs. In the third stage, which involves downstream tasks, we only fine-tune the GNN while keeping the LLM frozen. Experimental results demonstrate that GAGA achieves a classification accuracy that is comparable to or even exceeds SOTA methods while requiring only 1% of the data to be annotated, making it $3 \sim 100$ times more time-and-cost-efficient. Our contributions can be summarized as follows:

- We propose GAGA, a lightweight framework for TAG representation learning. Compared to previous methods, GAGA considers an efficient strategy that only annotates representative nodes (edges), thereby reducing both the time and cost of annotation via LLMs. It recognizes and incorporates the topological relationships between these annotations by constructing an annotation graph.

- Unlike previous methods, which are based on fine-tuning the whole dataset, GAGA leverages the alignment instead. To achieve this, we align sub-annotation graphs with the sub-graphs in TAG. Specifically, we developed a two-level contrastive learning (sub-graph level and prototype level), which provides significant improvement in performance. These two components could also be used in other problems.

- We provide comprehensive experiments to demonstrate the performance of GAGA. Specifically, via experiments over six datasets, GAGA achieves a classification accuracy that is comparable to or even exceeds SOAT methods for node classification and link prediction tasks. Moreover, compared to previous LLM-augmented SOTA methods, GAGA only requires 1% of the data to be annotated, making it surprisingly efficient.

## 2 RELATED WORK

**Language Models for Text-Attributed Graph.** Traditional GNN approaches typically handle TAGs by converting textual attributes into features via shallow neural networks, such as bag-of-words, which limits the comprehensive understanding of textual semantics. Recent research has focused on embeddings based on language models (LMs) like BERT (Devlin, 2018) to address this issue. These fine-tuning pre-trained models can effectively generate deep embeddings to capture the rich semantic information within TAGs. There are two main architectures for empowering TAG tasks using LMs. The first is the cascaded architecture, where the textual information of nodes is independently encoded by LMs, and GNN models then aggregate the outputs to obtain the final embeddings, including TextGNN (Zhu et al., 2021a), GEAR (Zhou et al., 2019), GIANT (Chien et al., 2021), GPT-GNN (Hu et al., 2020b), and SimTeG (Duan et al., 2023). However, this approach separates text encoding from graph aggregation, which prevents a unified integration of the two processes. As a result, the nested architecture has also been widely studied. This approach integrates text encoding and graph aggregation, performing these tasks iteratively to better unify both processes. For example, Graphormer (Yang et al., 2021), GLEM (Zhao et al., 2022) and DRAGON (Yasunaga et al., 2022) follow this nested architecture.

LLMs such as ChatGPT (Brown et al., 2020) have shown tremendous potential in various NLP tasks. However, how to apply LLMs to graph-structured data, such as TAGs, remains a challenge (Chai et al., 2023; Qin et al., 2023). Chen et al. (2024) investigated the potential of LLMs in node classification tasks. Some works have already attempted to use LLMs for TAGs. He et al. (2023) proposed TAPE, which leverages LLMs to use the explanations generated during zero-shot classification as informative features for the graph. Another category of methods utilizes LLMs' strengths in text understanding to improve the graph's topology by enhancing the semantic understanding of node information. A representative example of this approach is the work by Sun et al. (2023), where they utilize LLMs to generate pseudo-labels and compute the semantic similarity between node attributes to either remove or add edges, thereby improving the graph's topology. Yu et al. (2023) propose ENG that enhances TAGs by using LLMs to generate additional nodes and the corresponding additional information. Additionally, Pan et al. (2024) employs a knowledge distillation approach to distill LLMs into graph models specifically for TAG learning. However, focusing solely on either nodes or edges has its limitations. Concentrating only on nodes (textual information) can result in the loss of the network's topological structure, while focusing exclusively on edges makes it difficult to fully capture the semantic information of the nodes themselves. Therefore, we need a structure that combines both nodes and edges to enhance the TAG representation, which is a strength of GAGA. Moreover, as mentioned above, all previous SOTA methods are time-consuming and costly, while our GAGA is highly lightweight and achieves almost the same performance.

**Explanation-Guided Learning.** With the advancement of AI, the importance of research in Explainable Artificial Intelligence (XAI) has become increasingly prominent. Although researchers have made significant progress in XAI techniques, leading to more attempts to generate explanations for Deep Neural Networks (DNNs), more profound issues, such as how to apply XAI techniques to improve the performance of DNN models, warrant further attention as research progresses (Ross et al., 2017; Gao et al., 2024). In the field of computer vision, there has been extensive research on

using explanation supervision to guide model training (Das et al., 2017; Linsley et al., 2018; Qiao et al., 2018; Mitsuhara et al., 2019; Zhang et al., 2019; Gao et al., 2022; Zhang et al., 2023; Saha & Roy, 2023). In NLP tasks, explanations are primarily generated through attention mechanisms or by utilizing auxiliary generative models (Bao et al., 2018; Strout et al., 2019; Zhong et al., 2019; Choi et al., 2020; Ghaeini et al., 2019; Jain et al., 2020; Stacey et al., 2022). Recently, several explanation supervision frameworks for GNNs have also emerged. For instance, GNES, proposed by Gao et al. (2021), optimizes model explanations and predictions through weak supervision and regularization of the model's explanations. Compared to these papers, in GAGA, we consider the annotation given by LLMs as an explanation, which can further guide us for contrastive learning.

## 3 PRELIMINARIES

**Text-Attributed Graph.** A Text-attributed Graph (TAG) can be formally represented as $G = (V, A, \{s_n\}_{n \in V})$, where $V$ is a set of $N$ nodes, $A \in \mathbb{R}^{N \times N}$ is the adjacency matrix, $s_n \in D^{L_n}$ is a sequential text associated with node $n \in V$, with $D$ as the dictionary of words or tokens, and $L_n$ as the sequence length.

**Large Language Models and Prompting.** Prompts can take various forms, such as a single sentence or longer paragraphs, and may include additional information or constraints to guide the model's behavior. Let $\mathcal{M}$ be an LLM that takes an input sequence $x = (x_1, x_2, \ldots, x_q)$ and produces an output sequence $y = (y_1, y_2, \ldots, y_m)$. The model is typically trained to optimize the conditional probability distribution $p(y|x)$, which assigns a probability to each possible output sequence $y$ given $x$. To incorporate a prompt $w$ with the input sequence $x$, we can concatenate them into a new sequence $\hat{x} = (w, x_1, x_2, \ldots, x_q)$. The conditional probability distribution $p(\hat{y}|\hat{x})$ is then computed using $\hat{x}$. Formally, the probability of the output sequence $y$ given $\hat{x}$ is:

$$p(\hat{y}|\hat{x}) = \prod_{i=1}^{m} p(y_i|y_{<i}, \hat{x}),$$

where $y_{<i}$ represents the prefix of the sequence $y$ up to position $i - 1$, and $p(y_i|y_{<i}, \hat{x})$ denotes the probability of generating $y_i$ given $y_{<i}$ and $\hat{x}$.

**GNNs for Node Classification and Link Prediction.** GNNs are utilized for both node classification and link prediction via leveraging the structural and feature information of graphs. In node classification, the objective is to assign labels to nodes based on their attributes and connections. This is achieved by updating each node's representation through a message-passing process defined by:

$$h_i^k = f^k \left( h_i^{k-1}, \text{AGG} \left( \{h_j^{k-1} : j \in \mathcal{N}_i\} \right) \right) \in \mathbb{R}^d,$$

where $h_i^k$ is the representation of node $i$ at layer $k$, $\mathcal{N}_i$ represents the set of neighbors, and $f^k$ is a neural network layer that integrates the previous layer's representation with aggregated information from neighbors via a function like sum.

In link prediction, the task is to predict the probability of an edge existing between two nodes, $i$ and $j$, based on their learned representations and the overall graph structure. This probability is modeled by $p(i, j) = f(h_i^k, h_j^k)$, where the function $f$ uses the representations derived from the node classification process to assess the likelihood between node $i$ and $j$.

## 4 GAGA: GRAPH ALIGNMENT GUIDED TEXT-ATTRIBUTED GRAPH LEARNING

In this section, we introduce our novel framework, GAGA. As illustrated in Figure 2, it primarily comprises three stages: annotation graph generation, two-level (subgraph) alignment, and downstream task fine-tuning. In the first stage, we first obtain a small but important set of nodes or edges through representative-based selection. Then, we generate annotations for these nodes or edges by prompting LLMs. These annotations are constructed into an annotation graph. In the second stage, we employ a two-level (subgraph level and prototype level) contrastive learning between the sub-annotation graphs and their corresponding sub-textual graphs, enabling lightweight fine-tuning of both the language model (LM) and GNNs using information from the selected nodes. In the third stage, which involves downstream tasks, we only fine-tune the GNN while keeping the LM frozen.

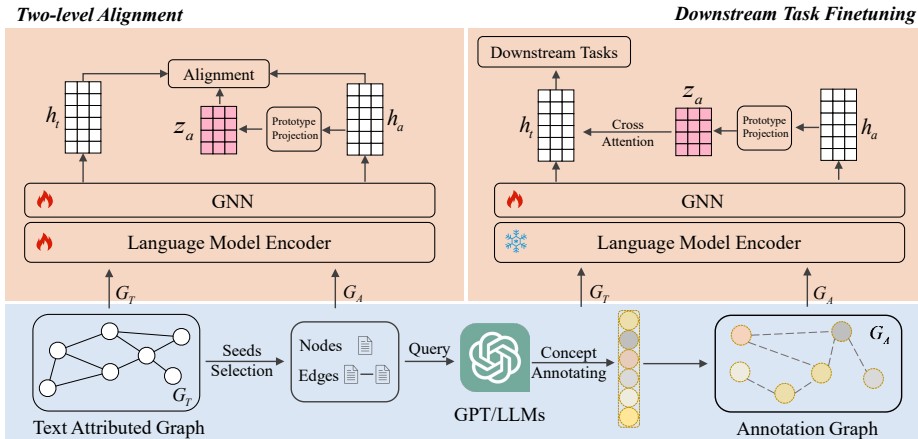

Figure 2: An overview of our GAGA framework.

## 4.1 ANNOTATION GRAPH GENERATION

The core idea of this stage is to use the information density metric to identify representative nodes and edges and then employ carefully designed prompts to utilize LLMs for annotation, thereby reducing costs.

**Representative Node (Edge) Selection.** As we mentioned earlier, using LLMs to annotate data can enhance model performance on TAGs. However, given that a typical graph contains tens of thousands of nodes, annotating each one can be both time-consuming and costly (Chen et al., 2023). Thus, to maintain high performance while annotating only a small number of nodes, it is crucial to select nodes (edges) with high information density (Cai et al., 2017). Information density, denoted as $\phi_{\text{density}}$, serves as a representative metric to identify nodes (edges) that best "represent" the underlying data distribution in the embedding space. To locate nodes situated in dense regions of the embedding space, we first perform $k$-means clustering on the embeddings of all unlabeled nodes. We then compute each node's Euclidean distance $||\cdot||$ to its nearest cluster center. The density score for a node $v_i$ is determined by converting this distance into a similarity score:

$$\phi_{\text{density}}(v_i) = \frac{1}{1 + ||\text{Emb}(v_i) - \text{Ce}(v_i)||},$$

where $\text{Emb}(v_i)$ represents the embedding of node $v_i$, and $\text{Ce}(v_i)$ denotes the center of the cluster to which node $v_i$ belongs. Intuitively, a higher value of $\phi_{\text{density}}(v_i)$ indicates that node $v_i$ is closer to its center, making it more representative in the embedding space. We denote the selected subset of nodes as $V_a^* \subset V$, which consists of the nodes with top-$s$ highest information density, where $s$ is a hyperparameter.

For link prediction tasks, we need to select a set of edges. For edge $e_{i,j}$, we define its information density as the sum of the information densities of the nodes at both endpoints: $\phi_{\text{density}}(e_{i,j}) = \phi_{\text{density}}(v_i) + \phi_{\text{density}}(v_j)$. Similar to nodes, the selection of the subset of edges, $V_a^*$ is based on the top-$\hat{k}$ highest information density.

**Annotation Generation.** Next, we will annotate these selected nodes (edges). In GAGA, we prompt LLMs not only to produce predictions and explanations for the selected nodes but also to infer and understand the broader concepts and knowledge needed for these predictions and explanations. For edges, we prompt LLMs to explain the rationale behind their formation and to grasp the concepts and knowledge necessary to understand these connections. Our prompt templates can be found in Appendix B.

**Annotation Graph Construction.** To structurally represent the most informative knowledge contained in each node's text and leverage potential relationships between annotations, we construct an annotation graph $G_{\mathcal{A}}^* = (V_a^*, E_a^*)$. In detail, we first define each annotation for nodes or edges as a node $v_a \in V_a^*$, and then we build the annotation graph based on the similarity between each pair of

nodes. For annotation node $v_a^i, v_a^j \in V_a^*$, the similarity $s(\cdot)$ is computed as follows:

$$s(v_a^i, v_a^j) = \frac{\text{Emb}(v_a^i) \cdot \text{Emb}(v_a^j)}{\|\text{Emb}(v_a^i)\| \cdot \|\text{Emb}(v_a^j)\|}.$$

We then apply the $k$-nearest neighbors (KNN) algorithm to capture the relationships between annotations, connecting each node to its $k'$ nearest neighbors with edges to form the annotation graph. Formally, the edges of the annotation graph $G_A^*$ is defined as:

$$E_a^* = \bigcup_i \{(v_a^i, v_a^j) \mid v_a^j \in N_{k'}(v_a^i)\},$$

where $N_{k'}(v_a^i)$ is set of $k'$-nearest neighbors of annotation node $v_a^i$.

## 4.2 TWO-LEVEL ALIGNMENT

As we have only annotated a subset of nodes (edges), how to generalize these annotations becomes the key question, which is the motivation of our two-level alignment. Our alignment consists of two levels: one is for aligning sub-textual graphs and sub-annotation graphs, while the other one is for aligning sub-textual graphs and the prototypes induced by sub-annotation graphs.

**Subgraph Alignment.** To adapt the Language Model Encoder for graph tasks, we perform graph alignment. Given that only a subset of nodes (edges) are annotated, we have to sample and align subgraphs based on this subset during this alignment process for better generalization. To achieve this, we use a straightforward approach: for each selected node $v^*$, we sample its $k$-hop neighbors in both the TAGs and annotation graphs to form a sub-text graph $G_T$ and sub-annotation graph $G_C$, respectively. For selected edges $e^*$, the difference lies in including the $t$-hop neighbors of both nodes connected by the selected edges to form the sub-annotation graph.

Through subsampling, we can obtain pairs of $G_T$ and $G_A$ corresponding to each selected node (or edge). Our alignment goal is to maximize the similarity between each pair of $G_T$ and $G_A$, while simultaneously minimizing the similarity between non-corresponding pairs. Specifically, for each graph, nodes obtain their representations through a Language Model Encoder (LMEncoder), which are then integrated with graph structural information via a GNN. Mathematically, we have the following loss:

$$\mathcal{L}_1 = \sum_i (\|h_t^i - h_a^i\|^2 - \frac{1}{n-1} \sum_{j \neq i} \|h_t^i - h_a^j\|^2), \tag{1}$$

$$h_T = \text{GNN}(X_T), h_A = \text{GNN}(X_A),$$

$$X_T = \text{LMEncoder}(G_T), X_A = \text{LMEncoder}(G_A),$$

where $h_T = \{h_t^1, h_t^2, \ldots, h_t^n\}$ and $h_A = \{h_a^1, h_a^2, \ldots, h_a^n\}$ is the embedding matrix for $G_T$ and $G_A$ with number of nodes $n$ in these subgraphs.

**Prototype Alignment.** In practice, we find the embedding matrix $h_A$ for the annotation graph is quite large, which makes the computational complexity high (linear in $n$) for downstream tasks. Additionally, there can be a significant amount of redundant information in $h_A$. By using more compact representations, the performance of downstream tasks may be effectively enhanced. To address these issues, we employ vector quantization (VQ) (Van Den Oord et al., 2017) for projection to get prototypes of the annotation graph representations. The prototype matrix is represented as a $k_p \times d$ matrix ($Z_a$), where $k_p$ denotes the size of the prototype projection, which is far less than $n$. Furthermore, the relatively small value of $k$ compels the model to learn more abstract representations of the annotations, thereby improving the generalization of the prototype embeddings. For the embedding vector $h_a$ in each annotation graph, its prototype $z_a$ is the closest vector in ($Z_a$):

$$z_a = Z_a^m, \quad \text{where} \quad m = \arg\min_j \left(\|h_a - Z_a^j\|_2\right), \tag{2}$$

where $Z_a^m$ is the $m$-th row vector of $Z_a^m$. After we have the prototype projections for each $h_a$, similar to (1), we can build another contrative loss for $h_t$ and $z_a$. The final loss function is the sum of these two, as expressed in the following equation.

$$\mathcal{L} = \alpha \sum_i \left[\|h_t^i - z_a^i\|^2 - \frac{1}{n-1} \sum_{j \neq i} \|h_t^i - z_a^j\|^2\right] + (1-\alpha)\mathcal{L}_1. \tag{3}$$

Here $\alpha$ is a hyper-parameter between 0 and 1, which controls the trade-off between subgraph textual information and prototype information during the model alignment process. Intuitively, $\alpha$ cannot be either too small or large. When $\alpha$ is small, the loss with more focus on the original contrastive loss (1), making generalization ability worse as the model will learn too much redundant information. When $\alpha$ is large, the model will focus more on prototype alignment, also making the utility worse as the project may lose too much information.

### 4.3 Downstream Task Finetuning

For a given downstream task, we first use the frozen Language Model Encoder to obtain the representations of nodes $X_T'$, which are then processed through GNN to acquire the node embeddings, i.e., $h = \text{GNN}(X_T')$. Finally, by computing cross attention with the prototype embeddings of the annotations, we integrate the information of the annotations and obtain the final node representation.

$$h' = \text{softmax}\left(\frac{hW^Q \cdot (Z_a W^K)^T}{\sqrt{d_k}}\right) \cdot Z_a W^V,$$

where $W^Q, W^K, W^V$ are query, key, $Z_a$ is the matrix of prototypes in (2), and value weight matrices and $d_k$ is the normalization factor.

For the node classification task, we add a fully connected layer and then pass the output through a softmax function to obtain the probability distribution for each category.

$$\hat{y} = \text{softmax}(f(h')),$$

where $f$ is a linear function. For the link prediction task, we use the element-wise multiplication of the representations of nodes $a$ and $b$ as the corresponding edge representation. This feature is then passed through a fully connected layer followed by a sigmoid function to obtain the probability of the edge's existence:

$$\hat{y} = \text{sigmoid}((h_a' \odot h_b') \cdot W).$$

## 5 Experiments

### 5.1 Experimental Setting

**Datasets.** We choose six datasets for evaluation: Cora (McCallum et al., 2000), PubMed (Sen et al., 2008), ogbn-arxiv (Hu et al., 2020a), ogbn-products (Hu et al., 2020a; He et al., 2023), and tape-arxiv23 (He et al., 2023) for the node classification tasks and use Cora, Citeseer (Giles et al., 1998b), and PubMed for the link prediction tasks. Details about the datasets can be found in Appendix A.1. The detailed prompts used for generating concepts for each dataset are shown in Appendix B. We split the datasets for training, validation, and testing following (He et al., 2023) for node classification tasks and HeaRT (Li et al., 2023; Mao et al., 2023) for link prediction tasks.

**Baselines.** In this study, we employed a variety of models for the node classification task to comprehensively evaluate the performance of different approaches on TAGs. Specifically, we selected the following eleven models: (i) MLP; (ii) GCN (Kipf & Welling, 2016a); (iii) GraphSAGE (Sun et al., 2021); (iv) RevGAT (Li et al., 2021); (v) InstructGLM (Ye et al., 2023); (vi) Graphormers (Ying et al., 2021); and (vii) TAPE (He et al., 2023) (viii) GLEM (Zhao et al., 2022) (ix) SimTEG (Duan et al., 2023) (x) ENGINE (Zhu et al., 2024) (xi) GIANT (Chien et al., 2021). Further details about the baselines are given in Appendix A.2.

For link prediction tasks, we selected four types of models, comprising a total of 19 models, as baselines: heuristic models, embedding models, GNN models, and GNN+Pairwise Info models. Heuristic models use graph structure-based scores to predict links. Embedding models learn low-dimensional representations of nodes to estimate link likelihoods, GNN models leverage multi-hop graph structures through message passing, and GNN+Pairwise Info models enhance GNNs with additional node-pair-specific information to improve link prediction accuracy. More details are given in Appendix A.2.

**Evaluation Metrics.** For node classification tasks, we use classification accuracy, time cost (min), and money cost of the whole framework as evaluation metrics. In the experiment of time and money comparison, time includes new textual feature annotation (like TAPE and GAGA), training and

inference time. Money usage only includes the money spent on annotating new features with LLMs. For link prediction tasks, we choose MRR@10 and AUC as metrics. See details in Appendix A.3.

**Experimental Setup.** In our experiments, we prompt GPT-3.5-turbo-1106 (Brown et al., 2020) to generate annotations. The Language Model Encoder utilized is all-MiniLM-L6-v2 (Wang et al., 2020), and the GNN employed is a 4-layer GCN. During the annotation step, we consider 40 clusters. For the node classification task, we use 1% nodes for annotation and $\sqrt{n_{\text{edges}}}$ for link prediction annotation with edge number $n_{\text{edges}}$. During the alignment process, we set $\alpha$ to 0.6, and prototype dimension $k_p$ to 40. We also use 2-hop subgraphs for the alignment. We use the Adam optimizer with the learning rate $5e^{-5}$ for alignment and $1e^{-3}$ for downstream task fine-tuning. All experiments were conducted on a 32G V100 card with 10 CPU cores and a maximum memory of 64G. We run each experiment for 5 times.

## 5.2 UTILITY ANALYSIS

**Time and Money Cost Comparison.** We present a comparative analysis of the previous LLM-based methods and GAGA on the ogbn-arxiv and PubMed datasets, focusing on time efficiency and cost efficiency. See Table 1 for results. From the results, we can easily see GAGA is highly efficient compared to previous methods. The main reason is that GAGA only used 1% of the nodes for annotation, whereas most other methods need annotating or fine-tuning all nodes. Due to the high API cost and large annotations, TAPE is very expensive compared to other models. We also observe that the ENGINE model runs out of memory. ENGINE cleverly combines LLM and GNN, and although it freezes the LLM weights to save memory, the Llama-3-7B used in the method still consumes a large amount of GPU memory, making it difficult to run on a single GPU when dealing with large datasets. As for the GIANT model, it uses a self-supervised approach to learn node representations, but its pre-training phase becomes extremely time-consuming as the number of nodes increases. We can also see that although GLEM and GraphFormers are more expensive than GAGA, they are the two most efficient methods among the previous methods. However, their accuracy is much lower than that of the SOTA results and GAGA.

Table 1: Comparison of Time (Minute) and Money Usage ($) Highlighted by Efficiency and Accuracy. OOM refers to instances where memory usage exceeded 32G of GPU memory or 64G of system memory. Timeout refers to tasks that remained incomplete after 72 hours. We highlight the best results in green .

| | ogbn-arxiv | | | PubMed | | |
|---|---|---|---|---|---|---|
| | Time Cost ↓ | Money Cost ↓ | Accuracy ↑ | Time Cost ↓ | Money Cost ↓ | Accuracy ↑ |
| TAPE | 694 | 113.39 | 75.20 | 71 | 17.63 | 94.31 |
| GLEM | 446 | 7.81 | 76.04 | 50 | 0.87 | 92.57 |
| SimTEG | 1439 | 25.18 | 75.13 | 283 | 4.95 | 81.23 |
| OneForAll | 185 | 3.24 | 0.6983 | 85 | 1.48 | 73.01 |
| ENGINE | OOM | OOM | OOM | 203 | 3.55 | 74.74 |
| InstructGLM | OOM | OOM | OOM | OOM | OOM | OOM |
| GraphFormers | 224 | 3.92 | 66.67 | 45 | 0.783 | 83.65 |
| GIANT | Timeout | Timeout | Timeout | 431 | 7.54 | 76.89 |
| GAGA(Ours) | 80 | 2.87 | 76.23 | 17 | 0.49 | 94.61 |

Table 2: Node classification performance comparison on different datasets (all values in %). The best results are highlighted with **bold** .

| Model | ogbn-arxiv | PubMed | Cora | ogbn-products | tape-arxiv23 |
|---|---|---|---|---|---|
| MLP | $53.36 \pm 0.15$ | $86.35 \pm 0.20$ | $63.88 \pm 0.12$ | $53.85 \pm 0.18$ | $62.02 \pm 0.25$ |
| GCN | $71.82 \pm 0.20$ | $80.31 \pm 0.15$ | $88.24 \pm 0.10$ | $70.52 \pm 0.12$ | $63.41 \pm 0.20$ |
| SAGE | $71.71 \pm 0.18$ | $88.81 \pm 0.15$ | $89.11 \pm 0.12$ | $69.13 \pm 0.20$ | $64.30 \pm 0.18$ |
| RevGAT | $70.83 \pm 0.15$ | $88.50 \pm 0.20$ | $89.11 \pm 0.10$ | $69.64 \pm 0.15$ | $65.63 \pm 0.20$ |
| Graphormer | $72.81 \pm 0.18$ | $88.24 \pm 0.15$ | $80.41 \pm 0.20$ | $68.15 \pm 0.12$ | $62.87 \pm 0.25$ |
| InstructGLM | $75.70 \pm 0.20$ | $93.84 \pm 0.15$ | $87.08 \pm 0.12$ | $65.32 \pm 0.02$ | $70.32 \pm 0.12$ |
| GLEM | $75.60 \pm 0.15$ | $92.57 \pm 0.20$ | $74.69 \pm 0.18$ | $73.77 \pm 0.15$ | $78.58 \pm 0.20$ |
| SimTEG | $75.29 \pm 0.18$ | $81.20 \pm 0.01$ | $88.04 \pm 0.12$ | $74.51 \pm 0.20$ | $79.51 \pm 0.15$ |
| ENGINE | $76.02 \pm 0.20$ | $74.72 \pm 0.15$ | $\mathbf{91.48 \pm 0.10}$ | $\mathbf{80.05 \pm 0.15}$ | $79.76 \pm 0.20$ |
| GIANT | $74.26 \pm 0.15$ | $76.90 \pm 0.20$ | $85.52 \pm 0.18$ | $74.06 \pm 0.12$ | $72.18 \pm 0.25$ |
| TAPE | $75.20 \pm 0.18$ | $94.31 \pm 0.15$ | $91.19 \pm 0.12$ | $79.96 \pm 0.20$ | $80.80 \pm 0.18$ |
| GAGA(Ours) | $\mathbf{76.21 \pm 0.15}$ | $\mathbf{94.62 \pm 0.20}$ | $89.67 \pm 0.12$ | $78.87 \pm 0.18$ | $\mathbf{81.03 \pm 0.25}$ |

Table 3: Link prediction results on Cora, Citeseer, and PubMed(%). The best results are highlighted with **bold** .

| | Models | Cora | | Citeseer | | PubMed | |
|---|---|---|---|---|---|---|---|
| | | MRR@10 | AUC | MRR@10 | AUC | MRR@10 | AUC |
| Heuristic | CN | 20.99 | 70.85 | 28.34 | 67.49 | 14.02 | 63.9 |
| | AA | 31.87 | 70.97 | 29.37 | 67.49 | 16.66 | 63.9 |
| | RA | 30.79 | 70.96 | 27.61 | 67.48 | 15.63 | 63.9 |
| | Shortest Path | 12.45 | 81.08 | 31.82 | 75.5 | 7.15 | 74.64 |
| | Katz | 27.4 | 81.17 | 38.16 | 75.37 | 21.44 | 74.86 |
| Embedding | Node2Vec | 37.29 ± 8.82 | 90.97 ± 0.64 | 44.33 ± 8.99 | 94.46 ± 0.59 | 34.61 ± 2.48 | 93.14 ± 0.18 |
| | MF | 14.29 ± 5.79 | 80.29 ± 2.26 | 24.80 ± 4.71 | 75.92 ± 3.25 | 19.29 ± 6.29 | 93.06 ± 0.43 |
| | MLP | 31.21 ± 7.90 | 95.32 ± 0.37 | 43.53 ± 7.26 | 94.45 ± 0.32 | 16.52 ± 4.14 | 98.34 ± 0.10 |
| GNN | GCN | 32.50 ± 6.87 | 95.01 ± 0.32 | 50.01 ± 6.04 | 95.89 ± 0.26 | 19.94 ± 4.24 | 89.69 ± 0.06 |
| | GAT | 31.86 ± 6.08 | 93.69 ± 0.27 | 48.69 ± 7.53 | 96.25 ± 0.20 | 18.63 ± 7.95 | 98.20 ± 0.07 |
| | SAGE | 37.83 ± 7.75 | 95.63 ± 0.27 | 47.84 ± 6.39 | 97.39 ± 0.15 | 22.74 ± 5.47 | 98.87 ± 0.04 |
| | GAE | 29.98 ± 3.21 | 95.08 ± 0.33 | 63.33 ± 3.14 | 97.06 ± 0.22 | 16.67 ± 0.19 | 97.47 ± 0.08 |
| GNN + Pairwise Info | SEAL | 26.69 ± 5.89 | 90.59 ± 0.75 | 39.36 ± 4.99 | 88.52 ± 1.40 | 38.06 ± 5.18 | 97.77 ± 0.40 |
| | BUDDY | 26.40 ± 4.40 | 95.06 ± 1.67 | 59.48 ± 8.96 | 96.72 ± 0.26 | 23.98 ± 5.11 | 98.2 ± 0.05 |
| | Neo-GNN | 22.65 ± 2.60 | 93.73 ± 0.36 | 53.97 ± 5.88 | 94.89 ± 0.60 | 31.45 ± 3.17 | 98.71 ± 0.05 |
| | NCN | 32.93 ± 3.83 | 96.76 ± 0.18 | 54.97 ± 6.03 | 97.04 ± 0.26 | 35.65 ± 4.60 | 98.98 ± 0.04 |
| | NCNC | 29.01 ± 3.83 | **96.90 ± 0.28** | 64.03 ± 3.67 | 97.65 ± 0.30 | 25.70 ± 4.48 | 99.14 ± 0.03 |
| | NBFNet | 37.69 ± 3.97 | 92.85 ± 0.17 | 38.17 ± 3.06 | 91.06 ± 0.15 | **44.73 ± 2.12** | 98.34 ± 0.02 |
| | PEG | 22.76 ± 1.84 | 94.46 ± 0.34 | 56.12 ± 6.62 | 96.15 ± 0.41 | 21.05 ± 2.85 | 96.97 ± 0.39 |
| | GAGA(Ours) | **46.22 ± 2.13** | 96.78 ± 0.02 | **64.83 ± 3.11** | **98.13 ± 0.11** | 44.31 ± 2.12 | **99.24 ± 0.01** |

**Node Classification.** In the node classification task, as shown in Table 2, GAGA significantly outperforms the previous classical GNN models, such as GCN, and transformer-based methods, such as Graphormer over all datasets. Moreover, compared to LLM-based methods, our method still performs quite well; it achieves the best for ogbn-arxiv, PubMed, and tape-arxiv23. While its performance on the Cora and ogbn-products datasets was second only to ENGINE and TAPE, the difference to the best, ENGINE, is less than 1.8%. These results demonstrate that GAGA achieves a classification accuracy that is comparable to or even exceeds SOTA methods due to the high generalization ability of our alignment method.

**Link Prediction.** Similarly, we conducted experimental evaluations on the link prediction task, whose results are shown in Table 3 in the Appendix. Thanks to the annotations on selected edges and the two-level alignment, GAGA almost achieves the best results across all evaluated datasets for both two metrics. It is only slightly less than NBFNet on the PubMed data for MRR@10.

## 5.3 ABLATION STUDY

In this section, we conducted ablation experiments on the validation and test sets of the ogbn-arxiv dataset for node classification tasks to investigate the impact of different configurations on our model. Additional studies can be found in Appendix Section D.

**Impact of the GNN Backbone on Performance.** Since our framework is plug-and-play and does not alter the structure of the GNN model, it can be used with any GNN backbone. We explored the impact of different GNN backbones on classification accuracy. We tested the effects on several different backbones, including MLP, GCN, SAGE, GAT, and RevGAT in Table 4.

While using various GNNs as backbones did affect the final classification accuracy of the model, the impact was not significant. The only exception is MLP; its accuracy is lower, maybe because it is too simple to capture the relationship between the nodes. This demonstrates the robustness of our model.

Table 4: Performance with different backbones on ogbn-arxiv.

| GNN | Test | Valid |
|---|---|---|
| MLP | 0.7287 | 0.7375 |
| GCN | 0.7621 | 0.7725 |
| SAGE | 0.7596 | 0.7651 |
| GAT | 0.7616 | 0.7704 |
| RevGAT | 0.7565 | 0.7597 |

Table 5: Performance with different ratio (%) of seed nodes on ogbn-arxiv.

| Ratio | Test | Valid |
|---|---|---|
| 0.1% | 0.7572 | 0.7701 |
| 0.2% | 0.7571 | 0.7700 |
| 0.4% | 0.7571 | 0.7699 |
| 0.6% | 0.7571 | 0.7699 |
| 0.8% | 0.7570 | 0.7700 |
| 1.0% | 0.7621 | 0.7725 |

**Impact of the Prototype Size $k_p$.** We used prototype embeddings to reduce the computational complexity during the downstream finetuning tasks. Here we will consider the effect of the prototype

dimension. In Table 6, we can see while increasing $k_p$ generally leads to higher computational costs, test accuracy remained relatively stable across different values. For instance, at $k_p = 10$, the model achieved a test accuracy of 0.7641 with the lowest time (200 seconds) and memory usage (5.07 GB). As $k_p$ increased to 1280, the accuracy slightly improved to 0.7672, but this came at the expense of significantly higher time (640 seconds) and memory (17.14 GB) requirements. Thus, the prototype projection is highly efficient since $k_p = 10$ already brings a good performance.

Table 6: Experimental results for different prototype dimension $k_p$ on ogbn-arxiv. Time and Memory are for the fine-tuning stage.

| $k_p$ | Time (s) | Memory (GB) | Test Accuracy |
|---|---|---|---|
| 10 | 200 | 5.07 | 0.7641 |
| 40 | 210 | 5.24 | 0.7666 |
| 80 | 223 | 5.47 | 0.7656 |
| 160 | 247 | 5.92 | 0.7646 |
| 320 | 298 | 7.44 | 0.7641 |
| 640 | 412 | 10.67 | 0.7656 |
| 1280 | 640 | 17.14 | 0.7672 |
| 1693 | 782 | 21.31 | 0.7666 |

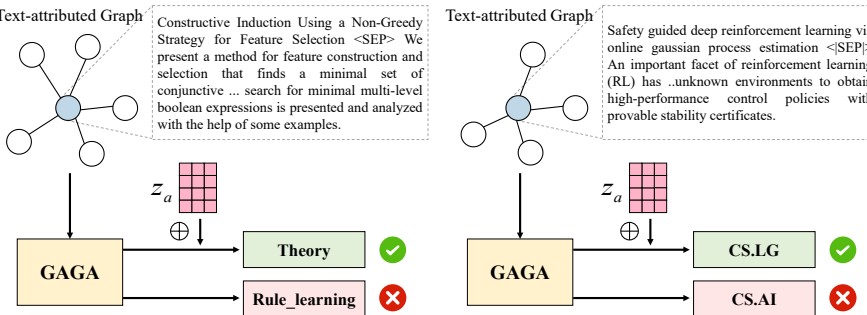

Figure 3: Visualization of the effect of annotation prototype projection.

**Impact of Number of Selected Nodes.** We study the effect of different proportions of selected nodes on performance. As shown in Table 5, while increasing the number of selected nodes can slightly improve classification accuracy, the model can already achieve strong performance (0.7641) even with just 0.1% annotated data. This efficiency is due to redundancy in the nodes' textual information. Specifically, in ogbn-arxiv, abstracts and titles of papers in the same category are highly similar, enabling effective learning from limited data. The use of high-information-density data allows the model to learn quality representations with fewer examples.

**Effect Visualization of Annotation Prototype.** We present two case studies, as shown in Figure 3, to illustrate how the annotation prototype enhances the model's performance. When the input textual information is processed by GAGA, directly predicting the category of the node can sometimes result in inaccurate predictions. However, by incorporating prototype information and integrating it into the model's prediction through a cross-attention mechanism, the model can effectively adjust and refine its predictions. This is because the integration allows the model to leverage additional contextual knowledge from the prototypes, thereby enhancing its ability to correct initial errors and produce more accurate final predictions. Additional visualization examples are in the Appendix C.

## 6 CONCLUSIONS

We investigated node classification and link prediction tasks in Text-attributed Graphs (TAGs) and introduced GAGA, a lightweight and highly efficient graph representation learning framework. GAGA employs a streamlined approach by annotating only representative nodes and edges, thereby significantly reducing both annotation time and cost. It leverages these annotations by constructing an annotation graph that captures the topological relationships among the annotated elements. Furthermore, GAGA features a two-level alignment module to effectively integrate the annotation graph with the TAG, ensuring the alignment of their underlying structures. Experimental results demonstrate that GAGA achieves classification accuracies comparable to or exceeding state-of-the-art methods while requiring only 1% of the data to be annotated, underscoring its high efficiency.

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

## A    ADDITIONAL EXPERIMENTAL DETAILS

### A.1    DATASETS

Table 7: Statistics of TAGs used in this work.

| Dataset | Nodes | Edges | Classes |
|---|---|---|---|
| Cora | 2,708 | 5,429 | 7 |
| PubMed | 19,717 | 44,338 | 3 |
| ogbn-arxiv | 169,343 | 1,166,243 | 40 |
| CiteSeer | 3,312 | 4,732 | 6 |
| ogbn-products | 54,025 | 74,420 | 47 |
| tape-arxiv23 | 46,198 | 78,543 | 40 |

The details of the datasets are as follows:

**Cora.** The Cora dataset (McCallum et al., 2000) consists of 2,708 scientific publications categorized into seven distinct classes: case-based, genetic algorithms, neural networks, probabilistic methods, reinforcement learning, rule learning, and theory. This dataset includes a citation network comprising 5,429 links. The selection process ensured that each paper in the final corpus either cites or is cited by at least one other paper.

**PubMed.** The PubMed dataset (Sen et al., 2008) comprises 19,717 scientific publications from the PubMed database, all related to diabetes. These publications are categorized into three distinct classes: Experimental Induced Diabetes, Type 1 Diabetes, and Type 2 Diabetes. The dataset also includes a citation network with 44,338 links.

**ogbn-arxiv.** The ogbn-arxiv dataset (Hu et al., 2020a) is a directed graph representing the citation network among all computer science papers on arXiv, as indexed by MAG (Wang et al., 2020). In this dataset, each node corresponds to an arXiv paper, and each directed edge signifies a citation from one paper to another. The primary task is to predict the 40 subject areas of the arXiv computer science papers, such as cs.AI, cs.LG, and cs.OS, which are manually assigned by the authors and arXiv moderators.

**Citeseer.** The CiteSeer dataset (Giles et al., 1998a) consists of 3312 scientific publications classified into one of six classes. The citation network consists of 4732 links. Each publication in the dataset is described by a 0/1-valued word vector indicating the absence/presence of the corresponding word from the dictionary. The dictionary consists of 3703 unique words.

**ogbn-products** The ogbn-products Hu et al. (2020a) dataset is an Amazon product co-purchasing network with products as nodes and edges indicating co-purchases. The task is to predict the product category in a multi-class setup using 47 top-level categories as labels.

**tape-arxiv23** The tape-arxiv23 He et al. (2023) dataset is a directed graph of citation networks among computer science arXiv papers from 2023 onwards. Each node represents a paper, and the edges show citation links. The task is to predict the 40 subject areas of these papers, such as cs.AI, cs.LG, and cs.OS, based on author and moderator labels.

### A.2    BASELINES

Details about the baseline models for node classification are as follows:

**MLP.** As a baseline model, MLP does not consider graph structure information and relies solely on node features for classification. Its performance provides a reference point for assessing the improvements offered by other models.

**GCN.** GCN (Kipf & Welling, 2016a) is a classical graph neural network model that effectively captures local neighborhood information through graph convolution operations. We chose GCN due to its strong performance and widespread application across many graph datasets.

**GraphSAGE.** GraphSAGE (Sun et al., 2021) generates node embeddings by sampling and aggregating information from neighboring nodes, making it suitable for handling large-scale graph data. We used GraphSAGE to evaluate its generalization capability across different datasets.

**RevGAT.** RevGAT (Li et al., 2021) combines graph attention mechanisms with reversible network structures, reducing memory consumption while maintaining performance. We selected RevGAT to explore the effectiveness of attention mechanisms on graph data.

**InstructGLM.** InstructGLM Ye et al. (2023) integrates the strengths of graph neural networks and language models, enhancing generalization through instruction learning. We here use InstructGLM using Llama-7b (Touvron et al., 2023) as the backbone.

**Graphormer.** Graphormer (Ying et al., 2021) leverages the Transformer architecture to process graph data, capturing global dependencies. We selected Graphormer to test the performance of Transformer-based approaches in graph neural networks.

**TAPE.** The TAPE model (He et al., 2023) leverages LLMs to capture textual information as features, which can be used to boost GNN performance on downstream tasks. A key innovation is its use of explanations as features: TAPE prompts an LLM to perform zero-shot classification, requests textual explanations for its decision-making process, and designs an LLM-to-LM interpreter to translate these explanations into informative features for downstream GNNs. For fairness, we use TAPE with the backbone of the GCN model, which is the same as our model.

**GLEM.** GLEM (Zhao et al., 2022) combines graph models with LLMs, like DeBERTa [12], in a variational EM framework, alternating between updating the LLM and GNN in the E-step and M-step to enhance downstream task performance.

**OneForAll.** OneForAll (Liu et al., 2023) represents all nodes and edges as human-readable texts and encodes them from various domains into a unified space using LLMs. The framework then adapts to different tasks by adding task-specific prompts into the input graph.

**ENGINE.** ENGINE (Zhu et al., 2024) introduces a lightweight, tunable G-Ladder module to each LLM layer, using a message-passing mechanism to incorporate structural information. This allows token-level outputs from each LLM layer to pass through the G-Ladder, enhancing node representations for node classification.

**GIANT.** GIANT (Chien et al., 2021) utilizes XR-Transformers (Zhang et al., 2021a) for neighborhood prediction, producing an LLM that generates more effective feature vectors for node classification than bag-of-words and vanilla BERT embeddings.

**SimTEG.** SimTeG (Duan et al., 2023) fine-tunes a pre-trained LM with parameter-efficient methods for a task like node classification, then uses the LM's last hidden states as node embeddings for GNN training, significantly boosting performance on graph benchmarks.

For the link prediction tasks, we use the baseline models from HeaRT (Li et al., 2023; Mao et al., 2023).

**Heuristic methods**: Common Neighbor (CN) (Newman, 2001), Adamic Adar (AA) (Adamic & Adar, 2003), and Resource Allocation (RA) (Zhou et al., 2009) use common neighbors, while Shortest Path (Liben-Nowell & Kleinberg, 2003) and Katz (Katz, 1953) rely on path information to score link existence.

**Embedding methods**: Matrix Factorization (MF) (Menon & Elkan, 2011), Multilayer Perceptron (MLP), and Node2Vec (Grover & Leskovec, 2016) learn low-dimensional node embeddings to predict link likelihood.

**GNN methods**: Graph Convolutional Network (GCN) (Kipf & Welling, 2016a), Graph Attention Network (GAT) (Veličković et al., 2018), GraphSAGE (SAGE) (Sun et al., 2021), and Graph Autoencoder (GAE) (Kipf & Welling, 2016b) integrate multi-hop graph structures via message passing.

**GNN + Pairwise Information**: SEAL (Zhang et al., 2021b), BUDDY (Chamberlain et al., 2022), and Neural Bellman-Ford Network (NBFNet) (Zhu et al., 2021b) use subgraph features; Neo-GNN (Yun et al., 2021), Neighborhood Contrastive Network (NCN) (Wang et al., 2023), and NCNC (Wang et al., 2023) utilize common neighbor information, while Position Encoding Graph Neural Network (PEG) (Wang et al., 2022) employs positional encoding.

## A.3 EVALUATION METRICS

We use MRR@10 and AUC for link prediction tasks.

$$MRR = \frac{1}{N} \sum_{i=1}^{N} \frac{1}{\text{rank}_i},$$

where $N$ is the number of positive samples and $\text{rank}_i$ is the rank of the $i$-th positive sample.

$$AUC = \frac{1}{|\mathcal{D}^0| \times |\mathcal{D}^1|} \sum_{i \in \mathcal{D}^0} \sum_{j \in \mathcal{D}^1} \mathbb{I}(\text{score}_i > \text{score}_j),$$

where $\mathcal{D}^0$ is the set of positive samples, $\mathcal{D}^1$ is the set of negative samples, and $\mathbb{I}$ is the indicator function.

## B PROMPTS FOR GENERATING ANNOTATIONS

---

**Prompt Template**

{Node's textual information}

[Question]: Which of the following category does this node belong to: {possible categories}. Give 5 likely categories as a comma-separated list ordered from most to least likely.

List the most important concepts in the paper. And you should tell me what knowledge is needed to understand the concepts. After all, you should provide your reasoning.

[Your response]:

---

**Prompt for node classification on ogbn-arxiv.**

Abstract: {abstract}
Title: {title}
Question: Which arXiv CS subcategory does this paper belong to?
Give 5 likely arXiv CS sub-categories as a comma-separated list ordered from most to least likely, in the form "cs.XX", list the most important concepts in the paper.
And you should tell me what knowledge is needed to understand the concepts.
After all, you should provide your reasoning. Your response:

---

**Prompt for node classification on Arxiv-2023.**

Abstract: {abstract text}
Title: {title text}
Question: Which arXiv CS subcategory does this paper belong to? Give 5 likely arXiv CS sub-categories as a comma-separated list ordered from most to least likely, in the form "cs.XX", List the most important concepts in the paper.
And you should tell me what knowledge is needed to understand the concepts.
After all, you should provide your reasoning. Your response:

---

**Prompt for node classification on Cora.**

Abstract: {abstract text}
Title: {title text}
Question: Which of the following sub-categories of AI does this paper belong to: Case Based, Genetic Algorithms, Neural Networks, Probabilistic Methods, Reinforcement Learning, Rule Learning, Theory? If multiple options apply, provide a comma-separated list ordered from most to least related, then for each choice you gave, explain how it is present in the text. List the most important concepts in the paper.
And you should tell me what knowledge is needed to understand the concepts.
After all, you should provide your reasoning. Your response:

**Prompt for node classification on PubMed.**

Abstract: {abstract text}
Title: {title text}
Question: Does the paper involve any cases of Type 1 diabetes, Type 2 diabetes, or Experimentally induced diabetes? Please give one or more answers of either Type 1 diabetes, Type 2 diabetes, or Experimentally induced diabetes; if multiple options apply, provide a comma-separated list ordered from most to least related, then for each choice you gave, give a detailed explanation with quotes from the text explaining why it is related to the chosen option. List the most important concepts in the paper.
And you should tell me what knowledge is needed to understand the concepts.
After all, you should provide your reasoning. Your response:

**Prompt for node classification on ogbn-products**

Product description: {product description}
Question: Which of the following category does this product belong to: 1) Home & Kitchen, 2) Health & Personal Care, 3) Beauty, 4) Sports & Outdoors, 5) Books, 6) Patio, Lawn & Garden, 7) Toys & Games, 8) CDs & Vinyl, 9) Cell Phones & Accessories, 10) Grocery & Gourmet Food, 11) Arts, Crafts & Sewing, 12) Clothing, Shoes & Jewelry, 13) Electronics, 14) Movies & TV, 15) Software, 16) Video Games, 17) Automotive, 18) Pet Supplies, 19) Office Products, 20) Industrial & Scientific, 21) Musical Instruments, 22) Tools & Home Improvement, 23) Magazine Subscriptions, 24) Baby Products, 25) NAN, 26) Appliances, 27) Kitchen & Dining, 28) Collectibles & Fine Art, 29) All Beauty, 30) Luxury Beauty, 31) Amazon Fashion, 32) Computers, 33) All Electronics, 34) Purchase Circles, 35) MP3 Players & Accessories, 36) Gift Cards, 37) Office & School Supplies, 38) Home Improvement, 39) Camera & Photo, 40) GPS & Navigation, 41) Digital Music, 42) Car Electronics, 43) Baby, 44) Kindle Store, 45) Kindle Apps, 46) Furniture & Decor? Give 5 likely categories as a comma-separated list ordered from most to least likely, list the most important concepts. And you should tell me what knowledge is needed to understand the concepts.
After all, you should provide your reasoning. Your response:

**Prompt for link prediction.**

Paper 1: Title: {title1} Abstract: {abstract1} Paper 2: Title: {title2} Abstract: {abstract2}
Question: Why are these two papers related? list the most important concepts in the abstract. And you should tell me what knowledge is needed to understand the concepts. Based on the concepts, explain why they are related.
Your response:

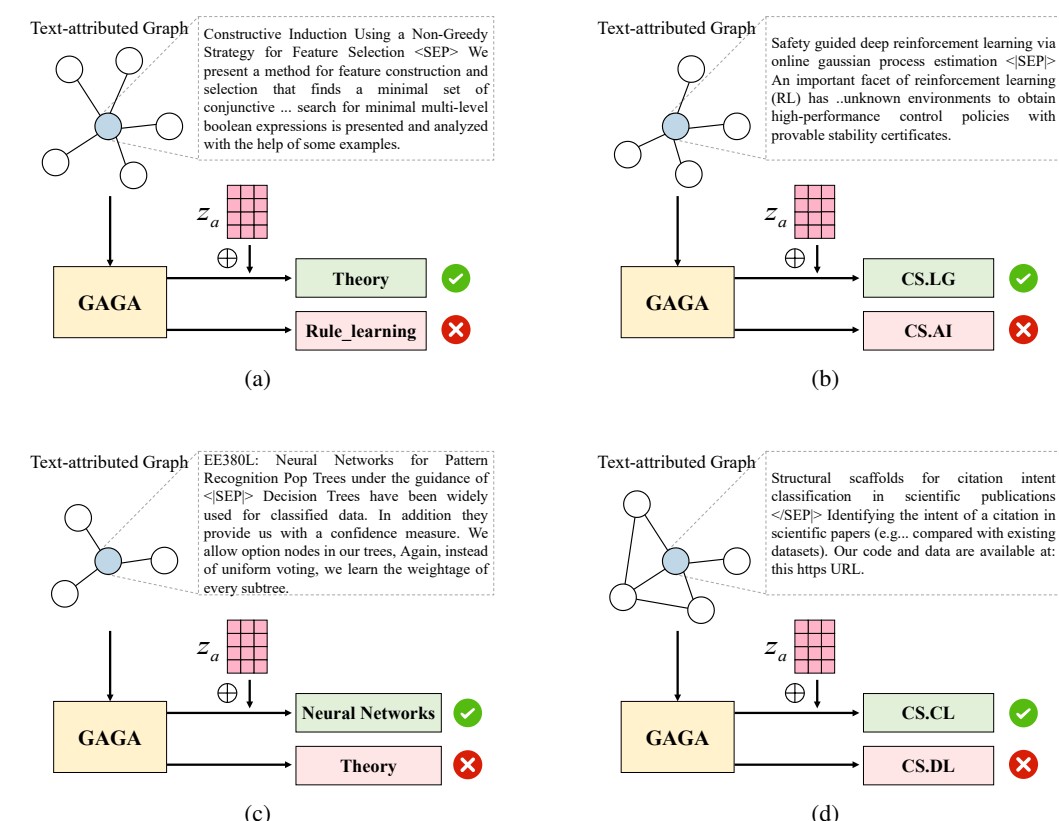

Figure 4: Visualization of the effect of annotation prototype projection.

## C    EFFECT VISUALIZATION OF ANNOTATION PROTOTYPE

In this section, we present several case studies, as shown in Figure 4, to illustrate how the annotation prototype enhances the model's performance. When the input textual information is processed by GAGA, directly predicting the category of the node can sometimes result in inaccurate predictions. However, by incorporating prototype information and integrating it into the model's prediction through a cross-attention mechanism, the model can effectively adjust and refine its predictions. This is because the integration allows the model to leverage additional contextual knowledge from the prototypes, thereby enhancing its ability to correct initial errors and produce more accurate final predictions.

## D    ADDITIONAL ABLATION STUDY

**Impact of Text Encoder.** In this section, we aim to investigate the impact of different language models on the performance of GAGA. We replace the language model with various alternatives and compare their performance on the ogbn-arxiv dataset. In GAGA, we align annotations and graphs, both of which are represented by features extracted from a language model. Therefore, the selection of an appropriate language model significantly impacts the final performance of the model. We compared the effects of different language models on the overall classification accuracy. The results in Table 8 showed that using larger models may not improve classification accuracy. This may be due to the high dimensionality of embeddings in larger models, which may hinder the learning process of the GNN. Furthermore, due to the limited data utilized during the alignment process, it is possible that larger language models were not adequately aligned, contributing to the suboptimal performance observed.

Table 8: Ablation study on the effects of different LLMs for annotation.

| Backbone | all-MiniLM-L6-v2 | MiniLM-L12-H384-uncased | gte-Qwen2-1.5B-instruct | gte-Qwen2-7B-instruct | LLM2Vec-Meta-Llama-3-supervised |
|---|---|---|---|---|---|
| Model Size | 23M | 33M | 1.5B | 7B | 8B |
| Test | 0.7620 | 0.7681 | 0.7253 | 0.7515 | 0.7519 |
| Valid | 0.7701 | 0.7722 | 0.7404 | 0.7531 | 0.7556 |

**Impact of** $\alpha$. In Figure 5, we studied the effect of the hyperparameter $\alpha$ in equation (3), which adjusts the model's focus between the alignment of sub annotation graph and sub textual graph, and the alignment of sub textual graph and prototype projections. As shown in the figure, when $\alpha$ is set to 0, the model disregards the prototypes and focuses on the original contrastive loss, making generalization ability worse as the model will learn too much redundant information. When $\alpha$ is set to 1, it focuses more on prototype alignment and ignores the original sub annotation graphs, making the utility worse as the project may lose too much information. Both extreme cases result in a decline in model performance. However, when $\alpha$ is between 0 and 1, the model's performance remains relatively stable.

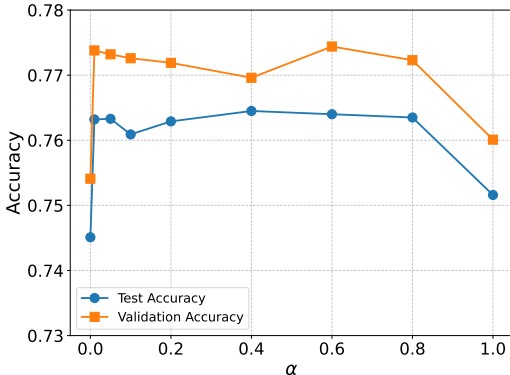

Figure 5: Impact of $\alpha$ on Valid and Test Accuracy on ogbn-arxiv Dataset.

