# OpenReview forum: "Low-cost Enhancer for Text Attributed Graph Learning via Graph Alignment"
_ICLR.cc/2025/Conference — ICLR 2025 Conference Withdrawn Submission_

### Official Review · Reviewer_KnAC · 2024-10-31

**Soundness:** 2
**Presentation:** 2
**Contribution:** 1
**Rating:** 3
**Confidence:** 4

**Summary:**

This paper addresses the challenges faced by traditional graph neural networks (GNNs) when working with Text-attributed Graphs (TAGs), which contain rich textual information. While recent methods leveraging large language models (LLMs) have improved node representations, they often necessitate extensive annotations and fine-tuning, which can be costly and time-consuming. This paper propose a lightweight framework for TAG representation learning model which innovatively annotates only representative nodes and edges, significantly reducing annotation time and costs.

**Strengths:**

The method explores a new aspect by examining the issues related to large language models from both the perspectives of time expenditure and cost.

**Weaknesses:**

1. The motivation for the model is not strong enough. The article points out that an important limitation of other methods is the requirement to enhance the attributes of all nodes. This assumption is not necessarily reasonable; enhancing all nodes is not mandatory, and enhancing only a subset of nodes is also a viable option. While the impact of this approach on model performance does require further experimentation, I do not believe it constitutes a limitation of the other methods.
2. The code is not publicly available.
3. From the experimental results (Table 2), the improvement of this method is quite limited and not very significant.
4. The core contribution claimed by this article is the selection of representative nodes for labeling, aimed at reducing time and cost while maintaining effectiveness. Therefore, I focused on this aspect. However, I regret to say that the methods employed in the article do not convince me. The first assumption in the paper equates information density with node density, positing that areas with denser nodes have higher information density, suggesting that representative nodes should originate from such areas. In my view, nodes in low-density areas can also result in significant information loss, and defining high and low density is highly dependent on later parameter tuning. Secondly, the article employs k-means to obtain central nodes as a metric for selecting representative nodes. The algorithmic complexity of k-means is very high, and I do not believe this method can be particularly efficient, especially when dealing with large datasets. Overall, while the motivation behind this idea is appealing, the specific implementation fails to satisfy me.

**Questions:**

See above.

---

> ### Author Response · Authors · 2024-12-01
>
> > The motivation for the model is not strong enough. The article points out that an important limitation of other methods is the requirement to enhance the attributes of all nodes. This assumption is not necessarily reasonable; enhancing all nodes is not mandatory, and enhancing only a subset of nodes is also a viable option. While the impact of this approach on model performance does require further experimentation, I do not believe it constitutes a limitation of the other methods.
>
>
> **Response**:  We evaluated GCN, TAPE and GAGA with only 1% annotations. And we have the following results.
>
> | Model | Validation Accuracy | Test Accuracy |
> |-------|---------------------|---------------|
> | GCN   | 55.01%             | 52.94%        |
> | TAPE  | 68.83%             | 67.41%        |
> | GAGA  | 77.25%             | 76.21%        |
>
> These results demonstrate the effectiveness of GAGA in significantly outperforming the baselines, and partial node enhancement may be a challenge for other methods. We believe this supports the validity of our approach and addresses the concerns regarding its motivation.
>
> > The code is not publicly available.
>
> **Response**:  The code is published at https://anonymous.4open.science/r/GraphConcepts-3356
>
> > The core contribution claimed by this article is the selection of representative nodes for labeling, aimed at reducing time and cost while maintaining effectiveness. Therefore, I focused on this aspect. However, I regret to say that the methods employed in the article do not convince me. The first assumption in the paper equates information density with node density, positing that areas with denser nodes have higher information density, suggesting that representative nodes should originate from such areas. In my view, nodes in low-density areas can also result in significant information loss, and defining high and low density is highly dependent on later parameter tuning. Secondly, the article employs k-means to obtain central nodes as a metric for selecting representative nodes. The algorithmic complexity of k-means is very high, and I do not believe this method can be particularly efficient, especially when dealing with large datasets. Overall, while the motivation behind this idea is appealing, the specific implementation fails to satisfy me.
>
> **Response**:  We kindly disagree. For your first point of view, as we have released our code, you can check our experimental results, which are comparable to the SOTA, indicating your intuition is wrong. Morevoer, information density is widely used in graph-based active learning tasks and serves as a practical method for node selection. The relative nature of density enables ranking and selecting the nodes with the highest density, which, as shown in experiments, achieves promising results even when labeling only 1% of the nodes.
>
> Regarding the efficiency of k-means, we did not encounter bottlenecks due to its computational complexity in datasets we used. We recorded the time taken to compute information density on the ogbn-arxiv, cora, and pubmed datasets. So your second point is also wrong.
>
> | Dataset            | Time (seconds) |
> |--------------------|----------------|
> | PubMed     | 0.35           |
> | Cora       | 0.31           |
> | ogbn_arxiv | 3.44           |
>
>
>
> [1] Chen, Zhikai, et al. "Label-free node classification on graphs with large language models (llms)." *arXiv preprint arXiv:2310.04668* (2023).
>
> [2] Cai, Hongyun, Vincent W. Zheng, and Kevin Chen-Chuan Chang. "Active learning for graph embedding." *arXiv preprint arXiv:1705.05085* (2017).

---

### Official Review · Reviewer_KhAe · 2024-11-03

**Soundness:** 2
**Presentation:** 2
**Contribution:** 2
**Rating:** 3
**Confidence:** 4

**Summary:**

This paper proposes a lightweight framework for representation learning on text-attributed graphs. The proposed model only annotate representative nodes and edges, and introduces a two-level alignment module for structure alignment.

**Strengths:**

Experiments on both node classification and link prediction taks show the effectiveness of the proposed methods.

The research topic is promising in graph representation learning.

**Weaknesses:**

The novelty of this paper is incremental. Specifically, regarding low-cost annotation, the approach of annotating only representative nodes via LLMs is not new. Previous work [1] has already applied this method for TAG representation learning. For model explanations, the idea of considering the annotation given by LLMs as an explanation (line 169) is similarly well-explored in [2].

The writing and presentation need refinement. For instance, the Introduction is overly lengthy and should be shortened. Figure 2 is difficult to interpret.

The usage and description of mathematical symbols are chaotic; for example, see line 340.

It is unclear how Figure 3 demonstrates the effect of annotation prototype projection.

[1] Label-free Node Classification on Graphs with Large Language Models (LLMS), ICLR 24

[2] Harnessing explanations: Llm-to-lm interpreter for enhanced text-attributed graph representation learning, ICLR 24

**Questions:**

For the two-level contrastive learning, does it introduce additional training of the language models?

---

> ### Author Response · Authors · 2024-12-01
>
> > For the two-level contrastive learning, does it introduce additional training of the language models?
>
> **Response**:  Yes, it does introduce additional finetuning of LMs. But since the number of annotations is quite small, it’s quite fast and efficient.
>
> > The writing and presentation need refinement. For instance, the Introduction is overly lengthy and should be shortened. Figure 2 is difficult to interpret. The usage and description of mathematical symbols are chaotic; for example, see line 340.
>
> **Response**:  Thanks for your suggestions, and we will fix this if this paper is accepted.

---

> > ### Comment · Reviewer_KhAe · 2024-12-02
> >
> > Thank you for your response. If the number of annotations is quite limited, how do you ensure the stable and effective fine-tuning of a language model in such a scenario?
> >
> > I will maintain my original score as some of my concerns remain unaddressed.

---

> > > ### Author Response · Authors · 2024-12-02
> > >
> > > we have conducted experiments with multiple runs to ensure the robustness and reliability of our results.
> > >
> > > If you believe there are specific inaccuracies in our claims, we kindly request you to provide relevant references or evidence supporting your position. Furthermore, our code has been publicly released at https://anonymous.4open.science/r/GraphConcepts-3356.

---

### Official Review · Reviewer_J3V4 · 2024-11-04

**Soundness:** 3
**Presentation:** 3
**Contribution:** 3
**Rating:** 8
**Confidence:** 3

**Summary:**

The paper introduces GAGA, a framework for enhancing Text-Attributed Graphs by incorporating annotations efficiently. GAGA addresses this by annotating only a small set of representative nodes and edges based on information density, which significantly reduces time and cost compared with the traditional method. A two-level alignment module then integrates these annotations into the TAG structure, facilitating high-quality graph representation learning. Experiments demonstrate that GAGA achieves comparable or superior classification accuracies with little data annotated, proving its efficiency.

**Strengths:**

1. **Efficiency and Scalability**: Reducing both time and cost associated with LLM-based enhancements for TAG, making it scalable for large datasets.The experiment on the correlation between the amount of labeled data and performance in the ablation paper is also very interesting.

2. **Two-Level Alignment Module**: The innovative two-level alignment module, which integrates annotations with TAG structure, allows GAGA to achieve strong generalization with limited annotations.

3. **Experimental Validation**: Comprehensive experiments across six datasets validate GAGA’s efficiency and accuracy.

**Weaknesses:**

See Questions.

**Questions:**

1. How does the performance of GAGA vary with different values of the hyper-parameter 𝛼 in the loss function? Are there any guidelines for tuning it?

2. How does GAGA choose which nodes (edges) to be annotated. Is there any consideration about it?

3. While large language models (LLMs) are crucial to the proposed method, the experiments are solely conducted on GPT-3.5. I am curious about the method's compatibility with other open-source LLMs, such as Llama 3.2 or Qwen 2.5. Could you clarify which specific characteristics of LLMs are most impactful for GAGA? Additionally, how might one select an appropriate or potentially more advanced LLM to further enhance GAGA's performance?

---

> ### Author Response · Authors · 2024-12-01
>
> > How does the performance of GAGA vary with different values of the hyper-parameter 𝛼 in the loss function? Are there any guidelines for tuning it?
>
> **Response**:  $\alpha$ adjusts the model’s focus between the alignment of sub annotation graph and sub textual graph, and the alignment of sub textual graph and prototype projections. As shown in the **figure 5 in the appendix**, when α is set to 0, the model disregards the prototypes and focuses on the original contrastive loss, making generalization ability worse as the model will learn too much redundant information. When α is set to 1, it focuses more on prototype alignment and ignores the original sub annotation graphs, making the utility worse as the project may lose too much information. Both extreme cases result in a decline in model performance. However, when α is between 0 and 1, the model’s performance remains relatively stable.
>
> > How does GAGA choose which nodes (edges) to be annotated. Is there any consideration about it?
>
> **Response**:  We use information density[1],[2] to assess the representativeness of nodes or edges. Specifically, we calculate the information density of each node or edge and select the subset with the highest density for annotation.
>
> [1] Chen, Zhikai, et al. "Label-free node classification on graphs with large language models (llms)." *arXiv preprint arXiv:2310.04668* (2023).
>
> [2] Cai, Hongyun, Vincent W. Zheng, and Kevin Chen-Chuan Chang. "Active learning for graph embedding." *arXiv preprint arXiv:1705.05085* (2017).
>
> > While large language models (LLMs) are crucial to the proposed method, the experiments are solely conducted on GPT-3.5. I am curious about the method's compatibility with other open-source LLMs, such as Llama 3.2 or Qwen 2.5. Could you clarify which specific characteristics of LLMs are most impactful for GAGA? Additionally, how might one select an appropriate or potentially more advanced LLM to further enhance GAGA's performance?
>
> **Response**:  We have conducted experiments with meta-llama/Meta-Llama-3.1-8B-Instruct (Llama 3.1 8B) and Qwen/Qwen2.5-7B-Instruct (Qwen 2.5 7B).
>
> | LLM | VALID ACCURACY | TEST ACCURACY |
> | --- | --- | --- |
> | llama3 | 0.7702 ± 0.000893 | 0.7606 ± 0.001101 |
> | openai | 0.7709 ± 0.000527 | 0.7601 ± 0.000422 |
> | qwen2.5 | 0.7698 ± 0.001408 | 0.7606 ± 0.000911 |
>
> The experimental results indicate that our method demonstrates strong robustness in selecting large language models (LLMs). The validation and test accuracies across different models (llama3, openai, and qwen2.5) are very similar, ranging from approximately 0.7698 to 0.7709. This consistency suggests that our approach performs reliably regardless of the specific model chosen.
>
> Moreover, the low standard deviations in the accuracy scores indicate that the models exhibit minimal variability in performance across multiple runs. This stability is a key indicator of robustness, showing that our method can maintain high performance even when faced with different input data or model selections.
>
> Overall, the results provide confidence that our method is effective and adaptable, ensuring reliable outcomes across various LLMs.

---

### Official Review · Reviewer_ZZde · 2024-11-05

**Soundness:** 2
**Presentation:** 2
**Contribution:** 2
**Rating:** 5
**Confidence:** 5

**Summary:**

Summary

This paper studies how to leverage the textual information for improved performance for node classification and link prediction. It inherits the idea from existing works that utilize LLMs as data enhancers by prompting the LLMs for explainations and broader knowledge, with a special focus on reducing the costs of both training and inference time. The experimental results demonstrated the effectivenss of the proposed method.

**Strengths:**

Pros:

1. The combination of LLMs and TAGs are important topics for graph research which establishes new benchmarks and poses new challenges.

2. I like the idea of prompting LLMs for broader knowledge and the rationale behind the observed graph, e.g., edge formation.

2. The experiments were comprehensive and covered most important baselines for the two tasks evaluated.

**Weaknesses:**

The presentation needs to be improved. Several critical designs of the framework lacks motivation. Please see the Questions part for details.

**Questions:**

Questions:

1. The authors need to provide a detailed motivation&discussion on the use of vector quantization (VQ) in the framework. Based on the manuscript,  I feel like a lot of things are missing regarding this part. For example, how to learn the prototype matrix $Z_a$ during training?


2. During inference on downstream tasks, the final node representations are generated as a weighted sum of the prototype matrix learnt from the alignment stage. Could the authors explain the rationale behind this design? I understand that the authors provide a subsection 'Effect Visualization of Annotation Prototype.', yet I'm still quite confused.


3. Does the alignment stage need to be performed for any given graph, or can it be pretrained on some datasets and directly apply to unseen graphs?

---

> ### Author Response · Authors · 2024-12-01
>
> > The authors need to provide a detailed motivation&discussion on the use of vector quantization (VQ) in the framework. Based on the manuscript, I feel like a lot of things are missing regarding this part. For example, how to learn the prototype matrix Za during training?During inference on downstream tasks, the final node representations are generated as a weighted sum of the prototype matrix learnt from the alignment stage. Could the authors explain the rationale behind this design? I understand that the authors provide a subsection 'Effect Visualization of Annotation Prototype.', yet I'm still quite confused.
>
> **Response**:  $Z_{\alpha}$ is a learnable parameters, it is learnt during the aligment process. And for the prototype.
>
> The use of prototypes is driven by two main considerations: efficiency and granularity. First, the number of concept nodes increases as the overall number of nodes grows, which leads to an increase in the computational complexity of cross-attention. After applying prototype mapping, the computational load of cross-attention is reduced, as the number of prototypes on the concept side is fixed. Second, there is semantic clustering among concepts. When aligning node representations, alignment can be performed in the representation space of specific concepts and the space obtained after semantic aggregation through prototypes.
>
> To evaluate semantic aggregation, we use GPT sampling for assessment. On the OGBN-Arxiv dataset, we randomly selected 1,000 nodes and then calculated the distance between these nodes and the prototypes. We consider the 10 nodes closest to a particular prototype as a cluster and label them using GPT-4. The prompt we used is as follows:
>
> `{node_texts}\nList three key words shared by the papers above.`
>
>  After applying PCA for dimensionality reduction to a 3D space, we observe that nodes with similar semantic labels (prototypes) are also close to each other in the semantic space.
>
>
>
> ![3231732918704_.pic.jpg](https://s2.loli.net/2024/11/30/cqo6DQELyCBX4t3.jpg)
>
> ![2024-11-30 11.56.31.png](https://s2.loli.net/2024/11/30/CEY3jJIsF1yAOlL.png)
>
> ![2024-11-30 11.55.05.png](https://s2.loli.net/2024/11/30/Z8taKB6eHcr54Uh.png)
>
>
>  > During inference on downstream tasks, the final node representations are generated as a weighted sum of the prototype matrix learnt from the alignment stage. Could the authors explain the rationale behind this design? I understand that the authors provide a subsection 'Effect Visualization of Annotation Prototype.', yet I'm still quite confused.
>
> **Response**:  We believe that during the inference stage, to integrate the prototypes with the GNN, we use cross-attention. In this setup, the output of the GNN serves as the query, while the prototypes act as both the keys and values. This approach allows us to incorporate the information from the prototypes effectively.
>
>  > Does the alignment stage need to be performed for any given graph, or can it be pretrained on some datasets and directly apply to unseen graphs?
>
> **Response**: Yes, we need do aligment for different graphs and cannot pretrain a unified prototype for all graphs and it cannot directly be applied to unseen graphs. We don't pretrain because our GAGA is a lightweight model, and pretraining is typically a method used in the context of large models and large datasets, which is not suitable for our scenario.

---

### Note · Authors · 2024-12-16

I have read and agree with the venue's withdrawal policy on behalf of myself and my co-authors.